# Synthesis and Characteristic Valuation of a Thermoplastic Polyurethane Electrode Binder for In-Mold Coating

**DOI:** 10.3390/polym16030375

**Published:** 2024-01-30

**Authors:** Suk-Min Hong, Hyuck-Jin Kwon, Jung-Min Sun, Chil Won Lee

**Affiliations:** Department of Chemistry, College of Science and Technology, Dankook University, Cheonan 31116, Republic of Korea; 12190615@dankook.ac.kr (S.-M.H.); 12190616@dankook.ac.kr (H.-J.K.); glasofkd1@naver.com (J.-M.S.)

**Keywords:** polyurethane, in-mold coating, electrode binder, ethoxy isosorbide, polycarbonate diol

## Abstract

A polyurethane series (PHEI-PU) was prepared via a one-shot bulk polymerization method using hexamethylene diisocyanate (HDI), polycarbonate diol (PCD), and isosorbide derivatives (ISBD) as chain extenders. The mechanical properties were evaluated using a universal testing machine (UTM), and the thermal properties were evaluated using thermogravimetric analysis (TGA) and differential scanning calorimetry (DSC). The PHEI-PU series exhibited excellent mechanical properties with an average tensile strength of 44.71 MPa and an elongation at break of 190%. To verify the applicability of different proportions of PU as an electrode binder, PU and Ag flakes were mixed (30/70 wt%) and coated on PCT substrates, the electrodes were evaluated by four-point probe before and after 50% elongation, and the dispersion was evaluated by scanning electron microscopy (SEM). The electrical resistance change rate of PHEI-PU series was less than 20%, and a coating layer with well-dispersed silver flakes was confirmed even after stretching. Therefore, it exhibited excellent physical properties, heat resistance, and electrical resistance change rate, confirming its applicability as an electrode binder for in-mold coating.

## 1. Introduction

Injection molding is utilized in various industries such as the automotive and aerospace industries [1], and it requires a separate coating process to improve the physical properties of the substrate. However, since the process is complex and many defects occur, in-mold coating has been introduced to improve the process [2]. In-mold coating (IMC) is a technology in which liquid resin is injected into a mold, pressed, and then printed simultaneously with the molding, which has the advantages of an uncomplicated process, good transferability of the mold surface to reproduce fine patterns, and effective control of the thickness of the coating layer [3,4,5]. Electrode binders that can be three-dimensionally processed and molded into a curved surface with various curvatures and shapes, such as heating radiators for automotive interior trims, are being studied using IMC [6].

Previously, thermosetting resins were used to provide excellent mechanical properties by adjusting crosslinking agents and structural modification. However, cracks occurred on the surface of thermosetting resins owing to the three-dimensional structure, such as a curved surface and repeated external forces, thereby increasing the electrical resistance [7,8]. Hence, the development of thermoplastic resins that can withstand external forces along the substrate is crucial. However, electrode binders using thermoplastics have not been actively studied, and their application as electrode binders requires improvements in mechanical properties and heat resistance, as in-mold processing usually takes place at a temperature of 100–200 °C [9], which requires high heat resistance [10]. Polyurethane (PU) comprises soft and hard segments; hence, its mechanical and physical properties can be adjusted based on the monomer type [11,12]. PU can be used in various processes such as injection molding [13] and can be applied to different substrates because of its excellent mechanical properties and adhesion due to the strong hydrogen bonding of urethane bonds [14].

PU synthesis involves the urethane reaction between diols and diisocyanates [15]. Diols are soft segments, which are divided into polyether, polyester, and polycarbonate diols. Polycarbonate diols have excellent mechanical properties and weather, moisture, and heat resistance properties [16,17]. Owing to their high polarity, polycarbonate diols can also be used in coatings and binders [18,19]. Diisocyanates are hard segments that are used in PU production. Methylene diphenyl diisocyanate (MDI) and toluene diisocyanate (TDI) are known aromatic diisocyanates, which are widely used owing to their excellent mechanical properties. However, they suffer from problems such as a lack of flexibility and yellowing [20] and generate toxic aromatic amines during hydrolysis, which are harmful to the environment and are subject to many regulations [21,22]. Hexamethylene diisocyanate (HDI), an aliphatic diisocyanate has the advantage of its linear alkyl chain, which can provide flexibility. However, it lacks mechanical properties such as tensile strength and elasticity [23,24]. To address this problem, research is underway to improve the physical properties, heat resistance, and environmental issues of PU including isosorbide (IS) [25,26].

Isosorbide is a bio-based material and has excellent intermolecular chain packing owing to its chiral structure, which can result in an increase in heat resistance and mechanical strength [27,28]. However, improvements are needed due to its disadvantages such as a low reactivity due to secondary alcohols [29], resulting in reduced percent yields of products after synthesis [30], and poor properties due to the increased unreacted monomers of IS during polymerization [31]. Recent studies have been conducted to improve the reactivity of IS using isosorbide derivatives (ISBD) substituted with primary alcohols [32]. Conductive materials such as carbon nanotubes [33], graphene, and metal particles [34] are principally used to impart electrode properties to PU. Among these, metal particles are superior to organic conductors in terms of their conductivity [35,36]. Ag particles have a high electrical conductivity at room temperature, are less expensive than Au, and are more stable in air than Cu. Hence, different types of Ag particles have been explored [37]. Among these, Ag flakes (AGF) are known to have an excellent conductivity owing to their large contact area and nanometer scale thickness, thereby resulting in a high packing density [38,39,40].

In this study, a PHEI-PU series with improved heat resistance and mechanical properties was synthesized using ISBD, which has improved reactivity, as a chain extender. The synthesized PUs were characterized using chemical structure analysis to determine their chemical and mechanical properties. The electrical properties of the PU/AGF blends were confirmed using a four-point probe. Finally, the dispersibility and electrical resistance values after 50% elongation were measured using a universal testing machine (UTM) and scanning electron microscopy (SEM) to confirm their applicability as in-mold coatings. PU series based on thermoplastic resins have excellent mechanical properties and heat resistance, and PU/AGF blends have low electrical resistance and good dispersion, which can be used in various applications as in-mold coatings.

## 2. Materials and Methods

### 2.1. Materials

Polycarbonate diol (PCD) was sourced from Asahi Kasei Chemica (Tokyo, Japan), with a weight average molar mass of 2000. Novasorb EI (ISBD) was purchased from Samyang Innochem (Gunsan, Republic of Korea). HDI was obtained from Sigma Aldrich (St. Louis, MI, USA) and used as the hard segment. Dimethylformamide (DMF) and isopropyl alcohol (IPA) were obtained from Duksan Chemicals (Ansan, Republic of Korea) and used as the solvents for post reaction purification. To prepare the PU/AGF blends, HP0202F1T (LT Metal, Seoul, Republic of Korea) was used as the Ag flake. Ethyl carbitol acetate (ECA; Tokyo Chemistry Industry, Tokyo, Japan) was used as the urethane solvent in the formulation.

### 2.2. Synthesis of the PHEI-PU Series

Various PCD:ISBD ratios were used in the catalyzed one-shot bulk polymerization (Figure 1). PCD (59.5 g, 0.0297 mol) and ISBD (10.9 g, 0.0297 mol) were used as diols, with HDI (9 g, 0.0535 mol) as the diisocyanate. Reactant stoichiometry and precursor weights are provided as an example for PHEI-PU 3 in Table 1. Before the reaction, PCD and ISBD were dried in a vacuum oven at 90 °C for 6 h to remove the residual moisture. PCD and ISBD were dissolved in a 250 mL three-necked round bottom flask using a mechanical stirrer under a nitrogen atmosphere at 80 °C for 1 h. HDI was then added and stirred for 30 min. The mixture was then poured into a Teflon beaker and polymerized in an oven at 110 °C under a nitrogen atmosphere for 12 h to synthesize PU. Subsequently, the obtained PU was then dissolved in DMF at 20 wt% and reprecipitated in IPA. After purification, the product was vacuum dried in a vacuum oven at 50 °C for 24 h to completely remove the residual solvent. Finally, a PHEI-PU series of polyurethanes was obtained with an average yield of 93%.

### 2.3. Preparation of the PU/AGF Blend and Coating

The synthesized PHEI-PU series was dissolved at 30 wt% using ECA as the solvent at 80 °C. After cooling to room temperature, Ag flakes were added at 70 wt% and stirred at 200 RPM for 1 h at room temperature using a mechanical stirrer. After that, the PU/AGF blends were homogeneously mixed by ultrasonication for 2 h to prepare the PU/AGF blend series (Table 2).

The prepared PU/AGF blend series were prepared as shown in Table 2 and coated on a poly cyclohexylene dimethylene terephthalate (PCT) film to a thickness of 25 μm using an applicator and dried in an oven at 80 °C for 10 min.

### 2.4. Characterization of the PHEI-PU Series

To confirm the synthesis of the PHEI-PU series, measurements were made using gel permeation chromatography on a FUTECS NP4000 instrument (Futec, Seoul, Republic of Korea). Tetrahydrofuran (HPLC grade, Duksan, Ansan, Republic of Korea) was used as the eluent, with a flow rate of 1.0 mL/min, temperature of 40 °C, and concentration of 0.5 wt%. Polystyrene (17,500, 50,000, 128,900, 370,000, and 510,000 M_p_) was used as the standard.

The viscosity of the PHEI-PU series was measured using the small sample adapter SSA18/13R of the BROOK FIELD DVE model (BROOK FIELD Co., Ltd., Toronto, ON, Canada) with a viscosity range of 3–10,000 cP (MPa·s) and measuring volume of 6.67 mL. The PHEI-PU series was dissolved in ECA at 30 wt%, and the viscosity was measured at 25 °C.

Fourier transform infrared spectroscopy (FTIR; Spectrum two FTIR spectrometer, PerkinElmer, Waltham, MA, USA) was used to confirm the synthesis of PU and the presence of unreacted materials. The transmittance was measured in the wavelength range of 4000–650 cm^−1^ using the attenuated total reflection method.

Hydrogen nuclear magnetic resonance (^1^H-NMR) spectroscopy was performed using an AVANCE III 400 NMR instrument (Bruker, MA, USA) with CDCl_3_ as the solvent at a frequency range and sensitivity of 400 MHz and 220:1, respectively.

### 2.5. Mechanical Properties

The mechanical properties of the PHEI-PU series were evaluated using a Universal Testing Machine (Model 3344, Instron Engineering Corp., Canton, MA, USA), while its films were cut into dumbbell shapes of 0.2 mm × 5 mm × 50 mm (thickness × width × length). They were measured at an elongation rate of 100 mm/min^−1^ at room temperature, with the average value converted after 5 measurements.

### 2.6. Thermal Properties

The glass transition temperature of the PU was analyzed using differential scanning calorimetry (DSC; Exstar 7020, SEIKO, Tokyo, Japan) conducted in the temperature range of −70–250 °C (with a ramping rate of 10 °C/min) in two cycles using a Pt pan in a nitrogen atmosphere.

To evaluate the heat resistance, thermogravimetric analysis (TGA; TGA-50, Shimadzu, Kyoto, Japan) was performed in a Pt pan in a nitrogen atmosphere at a temperature range of 25–600 °C with a heating rate of 10 °C/min.

### 2.7. Electrical Properties

A UTM was used to stretch the PU/AGF blend series. The sample size was standardized to 1.5 × 10 cm. The resistivity of the prepared PU/AGF blends was measured using a four-point probe (T2001 A3-WD, Ossila, Sheffield, UK), and the thickness was calculated from the sheet resistivity, converted to resistivity, and averaged after five measurements. The thickness was measured using a thickness gauge (293-230-30, 547-315, mitutoyo, Sakado, Japan) and was determined by dividing the specimen into four sections and taking two measurements for each section. The resistivity formula was expressed as follows. The sheet resistance, which is the resistivity value for the area of a square plate, multiplied by the thickness of the sample, equals (1):Ω/□ = Ohm/square = Sheet resistanceSheet resistance × Thickness(cm) = Resistivity (Ω·cm)(1)

### 2.8. Surface Dispersion Measurement

The conductivity increases when the metal particles are well dispersed in the polymer. SEM (JSM-6510, JEOL Ltd., Tokyo, Japan) was used to measure the dispersion of silver flakes in the binder. Before the measurement, the samples were deposited using a Pt depositor at 30 mA for 90 s, the binder surface was measured at 1500× magnification using an energy of 5 kV, and the dispersion of Ag flakes before and after stretching by UTM was evaluated.

### 2.9. Adhesion Evaluation

To determine the adhesion between the substrate and the coated binder, a cross-cut adhesion test was performed using the Cross-cut adhesion test basic cutter (TQC Sheen, Capelle aan den Ijssel, The Netherlands). The evaluation method involves creating a grid pattern with the cutter on the coated specimen. The adhesion test tape (7.6 N/cm) is then applied, and the specimen is observed by tearing the tape off uniformly at an angle of 60 degrees within 5 min (Figure 1). The test is evaluated according to the EN-ISO 2409 standards [41], and the evaluation criteria are shown in Table 3.

## 3. Results and Discussion

### 3.1. Preparation of the PHEI-PU Series

The PHEI-PU series was synthesized via one-shot bulk polymerization. For reproducibility purposes, a total of three syntheses were performed and converted to average values. The GPC data of PHEI-PU series are shown in Table 4. The weight average molecular weight was 62,800, and the number average molecular weight was 40,000. Because the molecular weight of the PCD is larger than that of ISBD, the larger the PCD content, the larger the molecular weight. The polydispersity averaged 1.574, indicating similar polydispersity due to the improved reactivity with the primary alcohol in ISBD, which addressed the problems of IS [42]. The viscosity of the PHEI-PU series was found to average 4180 cP (mPa·s), and the viscosity data are shown in Table 5. The higher the PCD content, the greater the viscosity, because the hydrogen bonding and entanglement phenomenon between polymers increases as the molecular weight increases.

### 3.2. Characterization of PUs by ^1^H-NMR

To confirm the synthesis of the PHEI-PU series, 1H-NMR was performed on PHEI-PU 3, PCD, and ISBD, and the spectra are shown in Figure 2. The ^1^H-NMR data of the synthesized PHEI-PU 3 showed C-H peaks of alkyl groups of HDI and PCD at 1.3–2.0 ppm, N-CH peaks of urethane at 3.3 ppm, C-O peaks of bicyclic methylene proton of ISBD at 3.75–4.0 ppm, and C-O peaks in the ISBD at 4–4.8 ppm; these findings confirmed the synthesis of ISBD [43]. The N-H peak of secondary amine of urethane appeared at 7–8 ppm indicating the successful synthesis of PU.

### 3.3. Characterization of PUs by FTIR Spectroscopy

Figure 3 shows the FTIR spectrum of the PHEI-PU series. This was measured to identify the main peak of the PU. The C=O in the carbonate and urethane bonds identified asymmetric and symmetric stretching vibration peaks at 1738 cm^−1^ and 1676 cm^−1^, respectively [44], while the secondary amine in the urethane bond identified an N-H stretching vibration peak at 3356 cm^−1^ and a C-N stretching vibration peak and an N-H bending vibration peak at 1528 cm^−1^. It can be seen that the transmittance of the N-H bending vibration peak increases as the content of PCD decreases in the PHEI-PU series, which is due to the fact that the molecular weight of ISBD is smaller compared to the PCD, thus forming more urethane bonds; PHEI-PU 5 with a larger content of ISBD has the largest value [29]. We also saw a C-O peak at 1236 cm^−1^ and no isocyanate peak at 2250 cm^−1^ for all PHEI-PU series, confirming successful PU synthesis without any reactants [45].

### 3.4. Mechanical Properties

To comprehensively evaluate the physical characteristics of the PHEI-PU series, a series of elongation tests were conducted with the utilization of a Universal Testing Machine (UTM), as shown in Figure 4. The tensile strength and strain at break of the PHEI-PU series were measured to evaluate their mechanical properties. The tensile strength ranged from 27.53 to 60.50 MPa with an average of 44.71 MPa, and the strain at break ranged from 51.5% to 440% with an average of 190%, confirming excellent properties. The higher the content of PCD, the higher the strain at break and the lower the tensile strength, which is due to the lower glass transition temperature of PCD, which makes it more flexible. On the other hand, the higher content of ISBD increases the tensile strength due to the stiffness property caused by the chiral structure of ISBD, but it shows higher brittleness, which is attributed to the fracture of the coating film due to its inflexibility at room temperature, due to its high glass transition temperature [29,46]. The PHEI-PU series synthesized in this study showed superior mechanical properties compared to the thermoset PUs studied as electrode binders for in-mold binder coatings [8].

### 3.5. Thermal Properties

Figure 5 shows the DSC data for the PHEI-PU series. A low glass transition temperature improves the flexibility and is suitable as an in-mold binder because it is easy to process [38]. The average glass transition temperature of the PHEI-PU series was −26.1 °C. PHEI-PU 5 with a high ISBD content showed a large glass transition temperature due to the reduced intermolecular spacing caused by the excellent packing properties of the chiral ring structure of ISBD [47]. PHEI-PU 1, which has a large PCD content, shows a decrease in the glass transition temperature owing to the increased spacing between the molecules caused by the rotation of the linear structure of the PCD. The low glass transition temperature of the PHEI-PU series confirmed the applicability of PU as an in-mold binder.

TGA measurements were performed to validate the applicability of the PHEI-PU series for in-mold processing. Figure 6 shows the TGA data of the PHEI-PU series, confirming the two-step degradation behavior. At a 10% loss, the average temperature of the PHEI-PU series is 316 °C, with PHEI-PU 1 having the best thermal resistance. The primary degradation region was attributed to the degradation of PCD, as PCD decomposes between 280 and 400 °C [48]. As the content of PCD increases, the initial decomposition temperature increases, which is judged to be the reason for the good thermal resistance [49]. The average temperature of the PHEI-PU series at 90% loss was 383 °C. The secondary degradation region is the degradation region of ISBD, and the PHEI-PU 5 with a higher ISBD content was found to have better heat resistance due to the chiral ring structure of ISBD. These results confirm that all PHEI-PU series are thermally stable and can be applied to in-mold processes [9,37].

### 3.6. Electrical Properties

To be used as an in-mold electrode binder, the PU/AGF series must have a low rate of electrical resistance change under external forces generated by the curved coating. To evaluate the electrodes of the PU/AGF blends coated on the PCT films, 50% stretching was performed using a UTM (Figure 7, Table 6). The average resistivity of the PHEI-PU series before stretching was 8.26 × 10^−5^ Ωcm, and after stretching, it was 9.61 × 10^−5^ Ωcm. The average electrical resistivity change was found to be 14.24%, and the PHEI-PU 1 with a high PCD content showed the lowest resistivity and rate of electrical resistance, which is believed to be due to the rotational nature of the linear structure of PCD, which increases the space between the molecular structures, allowing the Ag flakes to penetrate well into the molecular structure and disperse well. All PHEI-PU series showed resistance changes within 20%, confirming their applicability as in-mold electrode binders.

### 3.7. Surface Dispersion Measurement

Surface SEM measurements were taken on the PU/AGF blends before and after stretching with UTM to check the dispersion of the silver flakes in the specimen cross section. Since Ag flakes are conductive with a large contact area, the better the dispersion, the greater the surface contact, and the lower the resistance. Figure 8 shows a scanning electron microscope image of the crosssection of a specimen before and after tensile testing of the PHEI-PU series. All specimens in the PHEI-PU series before and after tensioning show no cracks, and the silver flakes are well dispersed without clumping. In the case of PHEI-PU 1, the silver flakes are better connected and well dispersed, so we can expect a lower resistance value compared to the other series. Even for samples stretched by 50% using UTM, the surface photographs show a smooth crosssection without cracks. This confirms the good dispersion of the PHEI-PU series both before and after stretching.

### 3.8. Adhesion Evaluation

To measure the adhesion between the substrate and the electrode binder, the PU/AGF blend was coated on the PCT substrate, and a cross-cut adhesion test was performed (Figure 9). Poor adhesion between the substrate and binder in the process not only results in poor coating performance but also poor durability [7]. The adhesion tests showed that the PU/AGF blends adhered well to the substrate except for some removal on the sides. This is believed to be due to the polyurethane’s excellent adhesion properties to the substrate, and both PHEI-PU series confirmed good adhesion with less than 5% of the area removed, being Category 4B or higher of the ASTM adhesion test standard [50].

## 4. Conclusions

In this study, PU with improved heat resistance, mechanical properties, and reactivity was synthesized using ISBD as a chain extender in different proportions. To evaluate the electrode properties, PU/AGF blends were prepared using Ag flakes and coated on PCT substrates. The DSC showed an increase in the glass transition temperature with increasing ISBD content due to its chiral ring structure. UTM confirmed excellent mechanical properties with an average tensile strength of 44.71 MPa and an average strain at break of 190%, and the tensile strength increased with increasing ISBD content, but the strain at break decreased, which was attributed to the decreased flexibility caused by the high glass transition temperature. TGA confirmed good heat resistance with an average loss of 10% at 316 °C; the high degradation temperature of PHEI-PU 1 in the primary degradation zone was due to the degradation of PCD, while the high degradation zone of PHEI-PU 5 in the secondary degradation zone was due to the degradation of ISBD. Four-point probe resistance measurements confirmed the excellent electrode properties of the PHEI-PU series and found that the higher PCD content increases the space between molecules due to the rotational nature of the linear structure, which facilitates the penetration of metal particles, resulting in lower electrode resistance, and the electrical resistance change rates are all within 20%. In addition, SEM confirmed that none of the PHEI-PU series had surface cracks before and after stretching. Thus, the PHEI-PU series has been confirmed to be applicable as an in-mold type electrode binder in various fields such as films for automotive radiators and stretchable electronic materials due to its excellent dispersibility, heat resistance, and mechanical properties. In future research, we plan to further study the applicability of electrode binders for in-mold coatings by diversifying the flexible segments.

## Data Availability

Data are contained within the article.

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
