# Peer review of "Synthesis and Characteristic Valuation of a Thermoplastic Polyurethane Electrode Binder for In-Mold Coating"

_polymers, 2024, doi:10.3390/polym16030375_

Round 1

Reviewer 1 Report

Comments and Suggestions for Authors

The submitted manuscript named [Synthesis and characteristic evaluation of thermoplastic polyurethane electrode binder for in-mold coating] is not bad. Nevertheless, the authors should improve the language and clarify several issues before it can be published.

1) Abstract, line 20: I suppose that the term “resistance change rate” means “electrical resistance change rate”.

2) Page 1, lines 31-34: The sentence is unclear. Perhaps the authors wanted to say that the cracks DECREASE the resistance?

3) Page 2, line 54: The sentence looks strange. It seems to be incorrect grammatically. Please, rephrase.

4) Page 2, line 62: The authors speak about isosorbide molecule and then about its yield. They probably mean isosorbide after some polymerization, but this is not clear from the sentence. Please, improve the sentence.

5) Page 2, line 82: I guess that the term “dispersion” should be replaced by “dispersion of AGF”. Am I right? In any case, define the term so that it was clear.

6) Page 3, Scheme 1: The font size in the scheme is too low. The letters and numbers are hard to read. Please, increase the font size.

7) Page 3, section 2.2: The abbreviation AGF is not explained. From the context, one can guess that AGF = Ag flakes = silver flakes. Nevertheless, the authors use the abbreviation without previous definition. The same applies to the last paragraph of the Introduction.

8) Page 4, section 2.4, line 144: Correct the phrase: “The measurement conditions were measured…”

9) Page 4, Equation 1: The first line of Eq. 1 contains a strange symbol – an empty square. Is this correct? Even if such a non-standard symbol was correct, it should be explained and described in the text.

10) Page 7, section 3.4: It does not make sense to calculate average stress at break and strain at break. Instead, it should be fair to conclude that the addition of IBSD increased stress at break and decreased strain at break, i.e. it made the material stiffer, but more brittle.

11) Page 8, line 224: The sentence on line 224 is a nonsense. I suppose it should read something like: “TGA measurements were performed in order to verify the applicability of PHEI-PU series for the in-mold process.”

11) Page 9, Figure 6 (and the corresponding text in section 3.5: It is a nonsense to use two decimals for the temperatures determined from TGA curves. The precision of the measurement is definitely lower. I suppose that it is in units of degrees Celsius at best.

12) Page 10, lines 275-276: “Since flake-like Ag flakes…” sounds strange. Please correct.

13) Page 10, section 3.7: In fact, I do not understand what the authors mean by the term “surface dispersion”. Please, define the term clearly.

14) Page 11, Conclusions: The authors should claim clearly, that the increasing concentration of ISBD influenced the final properties both positively (such as increase in stress at break) and negatively (such as decrease in drawability = strain at break and flexibility = Tg).

Comments on the Quality of English Language

The language of the submitted manuscript is not completely bad. The text is quite understandable. However, there are numerous grammar mistakes throughout the whole text, which should be corrected. Please, revise the manuscript carefully and/or get it checked by someone more fluent in English.

Reviewer 2 Report

Comments and Suggestions for Authors

I consider that the manuscript are well written and presents results with good revelance to the field. I accept that the manuscript can be publish after minor revisions, considering some details that I consider missing.

Page 3, line 105: The polymerization was performed at 110 ºC, where? In a hot plate or inside an oven?

Page 3, linq 119: What kind of applicator did you used? How did you guarantee that the thickness was always the same?

Page 5, table 4 and 5: The presented values are an average of several samples or a value of a single sample? Did you confirmed the reproducibility of this results? 

Page 10, Figure 8: Did you performed EDS mapping to visualise the well distribution of Ag flakes?

Reviewer 3 Report

Comments and Suggestions for Authors

The manuscript from Hong et al is clearly structured and easy to follow when I read it. While the content is not among the most innovative ones in the field, the paper may be accepted if the authors can address the following questions.

1. In discussing the synthesis of PHEI-PU Series, please specify the stoichiometry of the reaction, i.e., the molar ratio of the total -NCO to the total -OH in the reaction system.

2. How the chiral ring structure of ISBD is related to the thermal and mechanical properties of the resulting PU?

3. When comparing the mechanical, thermal and electrical properties, please also show the properties of PU synthesized from PCD as the diol without any ISBD.

4. From Figure 6, it seems PU-1 which has the lowest ISBD ratio exhibits the best resistance to thermal degradation. How ISBD improves the heat resistance in this case?

5. In electrical measurement, is the rate of the stretching controlled and measured?

6. It would be interesting if you put the sample under oscillating stretching of a tiny strain (so the material is still within the elastic regime) for certain cycles, and then examine the change of conductivity. This can be done in DMA if the access is available.

Round 2

Reviewer 3 Report

Comments and Suggestions for Authors

The revision looks good to me and I'd recommend the publication of this article.